# Understanding Generalization of Preference Optimization Under Noisy Feedback

## Abstract

As large language models (LLMs) advance their capabilities, aligning these models with human preferences has become crucial. Preference optimization, which trains models to distinguish between preferred and non-preferred responses based on human feedback, has become a crucial component for aligning LLMs. However, most existing works assume noise-free feedback, which is unrealistic given the inherent errors and inconsistencies in human judgments. This paper addresses the impact of noisy feedback on preference optimization, providing generalization guarantees under these conditions. Unlike traditional analyses that assume convergence, our work focuses on finite-step preference optimization, offering new insights that are more aligned with practical LLM training. We establish generalization guarantees for noisy preference learning under a broad family of preference optimization losses such as DPO, IPO, SLiC, etc. Our analysis provides the basis for a general model that closely describes how the generalization decays with the noise rate. Empirical validation on contemporary LLMs confirms the practical relevance of our findings, offering valuable insights for developing AI systems that align with human preferences.

## 1 Introduction

As large language models (LLMs) advance their capabilities, methods for aligning these models with human preferences have garnered significant research attention (Ji et al., 2023). Preference optimization, particularly through human-provided feedback, has emerged as a popular approach to ensuring that AI systems behave effectively and safely. A key recipe to achieve alignment is through the collection of binary preferences on generated outputs. In practice, human annotators are presented with two responses to the same question, and provide comparative judgments (*e.g.,* preferred, non-preferred) based on the quality of responses. Then, preference optimization algorithms such as those in Rafailov et al. (2023); Azar et al. (2023); Zhao et al. (2023); Tang et al. (2024) align the LLMs guided by the collected preferences. This process involves training models to assign a higher implicit reward to the preferred response over the non-preferred response. Preference-based alignment has demonstrated considerable success in enhancing the safety and usability of LLMs, making it a foundational component in the development of real-world LLM systems (OpenAI, 2023; Anthropic, 2023; Touvron et al., 2023; Gemini et al., 2023).

However, most existing works on preference optimization operate under the assumption of noise-free feedback. This assumption, while simplifying the problem, does not hold in real-world scenarios where human feedback is inherently noisy. The practical implications of noisy feedback are significant, as they directly impact the reliability and safety of AI systems deployed in critical applications. Factors such as human error, biases, and inconsistencies contribute to this noise, potentially leading to suboptimal or even harmful outcomes if not properly accounted for. Therefore, understanding the effects of noisy feedback in preference optimization is crucial for the development of robust, aligned AI systems. Recently, Gao et al. (2024b) empirically studied the impact of preference noise and observed that alignment performance can be sensitive to noise rates. However, *a rigorous theoretical understanding of these effects is still lacking*, underscoring the need for further research on this important problem.

In this work, we focus on the setting of noisy feedback in preference optimization and provide novel generalization guarantees under this condition. To the best of our knowledge, our results are the

first of their kind, addressing the gap in existing literature regarding the impact of noise on the generalization capabilities of preference learning algorithms. In particular, our theory is grounded in the context of *finite-step* preference optimization, which contrasts with classical learning theory literature assuming convergence or near-convergence of learning algorithms (Cao & Gu, 2020; Arora et al., 2019). By focusing on the finite-step setting, our analysis more accurately reflects the realities of LLM training, offering insights that are directly applicable to current practices of fine-tuning LLMs to avoid overfitting. This approach allows us to provide more realistic and practical guarantees for the generalization of preference optimization under noisy feedback, making our results particularly relevant for the development and deployment of robust AI systems.

In particular, we provide generalization guarantees for a broad family of preference optimization methods under noisy samples, encompassing existing algorithms such as DPO (Rafailov et al., 2023), IPO (Azar et al., 2023) and SLiC (Zhao et al., 2023) as special cases. All of these losses can be cast as a general form, referred to as generalized preference optimization (GPO) in Tang et al. (2024). Our guarantee captures how the generalization bound for GPO changes with the noise rate $\epsilon$, and based upon our theoretical results, we provide a general model that closely describes how the test error increases with the noise rate. The key insight of our **Theorem** 3.1 and **Theorem** 3.2 is that given the bound on the risk for when there is no noise, $\mathcal{R}_0$, we can determine an upper bound on the rate at which the risk increases with $\epsilon$. In particular, as $\epsilon$ increases from 0, the bound increases at a rate of $1/(1 - \sqrt{\mathcal{R}_0 \gamma} \epsilon)^2$, and as the noise rate approaches $1/2$, the expected risk transitions to growing at a linear rate. Our theory also reveals that stronger concentration, more samples, and contrasting directions for positive and negative samples yields tighter bounds and slower degradation in accuracy as the noise rate increases. We empirically verify our theory-based model on real-world dataset HH-RLHF (Bai et al., 2022a), demonstrating the practical relevance of our results. Overall, the close match between our theoretical analysis and empirical observation highlights the strength and applicability of our theoretical framework in modeling the effects of noise on preference optimization. Our contributions can be summarized as follows:

1. We establish the first generalization guarantees for preference optimization under noisy feedback. Our guarantees can be broadly applicable to *a generalized family of preference optimization approaches* (Tang et al., 2024), including DPO (Rafailov et al., 2023), IPO (Azar et al., 2023), SLiC (Zhao et al., 2023) as special cases.

2. We provide a comprehensive theoretical analysis of the impact of noise rate in the finite-step learning setting, leading to a general and practically relevant model that describes the effect of noise on generalization across various settings.

3. We conduct comprehensive empirical evaluations that support our theoretical findings and our derived model, showcasing the practical implications of our work.

## 2  PRELIMINARIES ON PREFERENCE OPTIMIZATION

We denote $\pi_\theta$ as a language model policy parameterized by $\theta$, which takes in an input prompt $x$, and outputs a discrete probability distribution $\pi_\theta(\cdot|x)$ over the vocabulary space $\mathcal{V}$. $\pi_\theta(y|x)$ refers to the model's probability of outputting response $y$ given input prompt $x$. Preference optimization typically operates on comparative data, where pairs of responses are presented, and the model is trained to discern the preferred choice. Formally, we define the preference data below.

**Definition 2.1 (Preference data).** *Consider two responses $y_w, y_l$ for an input prompt $x$, we denote $y_w \succ y_l$ if $y_w$ is preferred over $y_l$. We call $y_w$ the preferred response and $y_l$ the non-preferred response. Each triplet $(x, y_w, y_l)$ is referred to as a preference. Furthermore, the empirical dataset $\mathcal{D} = \{(x_i, y_{w,i}, y_{l,i})\}_{i=1}^N$ consists of $N$ such triplets sampled from a preference distribution.*

**Direct Preference Optimization (DPO).** To model the preferences, one popular framework is the Bradley-Terry model (Bradley & Terry, 1952), which assumes the following preference distribution

$$p^*(y_w \succ y_l|x) = \sigma(r^*(x, y_w) - r^*(x, y_l)), \tag{1}$$

where $\sigma$ is the logistic function and $r^*(x, y)$ is the reward function. The reward function takes in the prompt $x$ and response $y$ and outputs a higher scalar value $r^*(x, y)$ for the preferred response, and vice versa. Guided by Equation (1), one can learn a reward model either explicitly (i.e.,

by fitting a parametric reward model $r(x, y))$ or implicitly (i.e., via direct preference optimization (DPO) (Rafailov et al., 2023).

Explicit reward models are optimized to maximize the following binary classification objective:

$$\mathbb{E}_{(x,y_w,y_l)\in\mathcal{D}}[\log \sigma(r(x, y_w) - r(x, y_l))], \qquad (2)$$

which learns the reward function via maximum likelihood estimation (MLE) on the empirical preference dataset $\mathcal{D} = \{(x_i, y_{w,i}, y_{l,i})\}_{i=1}^N$, and $r$ is a function parameterized by a neural network. The resulting model is useful for RLHF (Christiano et al., 2017; Ouyang et al., 2022; Bai et al., 2022a; Ziegler et al., 2019), which aligns language models with the KL-constrained reward optimization:

$$\max_{\pi_\theta} \ \mathbb{E}_{\hat{y}\sim\pi_\theta(\cdot|x)}[r(x, \hat{y})] - \beta \log \frac{\pi_\theta(\hat{y}|x)}{\pi_{\text{ref}}(\hat{y}|x)}, \qquad (3)$$

where $\hat{y}$ is the output generated by the current model's policy $\pi_\theta$ for the prompt $x$, $\pi_{\text{ref}}$ is the policy of the model before any steps of RLHF, and $\beta$ is a regularization strength. We can view this objective as maximizing the expected reward with KL regularization weighted by $\beta$. We can see that the difference in reward is equivalent to the log ratio difference of the optimal policy to Equation (3):

$$r(x, y_w) - r(x, y_l) = \beta(\log \frac{\pi_\theta(y_w|x)}{\pi_{\text{ref}}(y_w|x)} - \log \frac{\pi_\theta(y_l|x)}{\pi_{\text{ref}}(y_l|x)}). \qquad (4)$$

DPO thus replaces the explicit reward function in Objective (2) with the implicit reward $r(x, y) = \log \frac{\pi_\theta(y|x)}{\pi_{\text{ref}}(y|x)}$, yielding the following objective to minimize:

$$\mathbb{E}_{(x,y_w,y_l)\in\mathcal{D}}\left[ - \log \sigma\left(\beta\left(\log \frac{\pi_\theta(y_w|x)}{\pi_{\text{ref}}(y_w|x)} - \log \frac{\pi_\theta(y_l|x)}{\pi_{\text{ref}}(y_l|x)}\right)\right)\right]. \qquad (5)$$

**Generalized Preference Optimization (GPO).** Recent work by Tang et al. (2024) presented a unified view of preference optimization encompassing existing algorithms including DPO (Rafailov et al., 2023), IPO (Azar et al., 2023) and SLiC (Zhao et al., 2023) as special cases. All of these losses can be cast as a general form, referred to as generalized preference optimization (GPO):

$$\mathbb{E}_{(x,y_w,y_l)\in\mathcal{D}}\left[f\left(\beta\left(\log \frac{\pi_\theta(y_w|x)}{\pi_{\text{ref}}(y_w|x)} - \log \frac{\pi_\theta(y_l|x)}{\pi_{\text{ref}}(y_l|x)}\right)\right)\right], \qquad (6)$$

where the function $f$ can be instantiated differently:

- DPO: $f(r_{\pi_\theta}(x, y_w, y_l)) = -\log \sigma(r_{\pi_\theta}(x, y_w, y_l))$ applies the logistic loss (Hastie et al., 2009).
- IPO: $f(r_{\pi_\theta}(x, y_w, y_l)) = (r_{\pi_\theta}(x, y_w, y_l) - 1)^2$ applies the squared loss (Azar et al., 2023).
- SLiC: $f(r_{\pi_\theta}(x, y_w, y_l)) = \max(0, 1 - r_{\pi_\theta}(x, y_w, y_l))$ applies the hinge loss function (Zhao et al., 2023).

In this paper, our theoretical analysis revolves around this **generalized formulation**, and thus can be broadly applicable to preference optimization losses in the GPO family. Specifically, we consider a set of objectives where $f(x)$ is a function with (i) $f'(0) < 0$ and $|f''(x)|$ bounded for all $x \geq 0$ or (ii) $f$ is the Hinge Loss as in SLiC. We define $D$ as $\sup_{x\geq0} |f''(x)|$ if $f$ satisfies (i) and we can set $D = \frac{1}{2\beta}$ for (ii).

## 3 GENERALIZATION OF GPO UNDER NOISY FEEDBACK

### 3.1 GENERALIZATION ANALYSIS TARGET

We begin by defining the analysis target for understanding the generalization behavior of preference optimization. From Equation (6), we can see that GPO learns to have a positive *reward margin* for a given sample $(x, y_w, y_l)$:

$$r_{\pi_\theta}(x, y_w, y_l) = \underbrace{\beta\left(\log \frac{\pi_\theta(y_{w,i}|x_i)}{\pi_\theta(y_{l,i}|x_i)} - \log \frac{\pi_{\text{ref}}(y_{w,i}|x_i)}{\pi_{\text{ref}}(y_{l,i}|x_i)}\right)}_{\text{Reward Margin}} > 0. \qquad (7)$$

Under the notion of reward margin, the population risk can also be defined formally below based on the notion of the reward margin.

**Definition 3.1** (**Population risk of preference learning**). *We define the population risk in terms of a 0-1 loss where a sample's loss is 0 when the reward margin is positive and 1 otherwise.*

$$\mathcal{R}(x, y_w, y_l) = \begin{cases} 0 & r_{\pi_\theta}(x, y_w, y_l) > 0 \\ 1 & r_{\pi_\theta}(x, y_w, y_l) \leq 0 \end{cases}$$

*where $r_{\pi_\theta}(x, y_w, y_l)$ is the reward margin for a new sample $(x, y_w, y_l)$. Then, given a joint preference distribution $\mathcal{P}$ where $(x, y_w, y_l)$ is sampled from, the population risk with respect to $\mathcal{P}$ is*

$$\mathcal{R}(\mathcal{P}) = \mathbb{E}_{(x, y_w, y_l) \sim \mathcal{P}} \left[ \mathcal{R}(x, y_w, y_l) \right]. \tag{8}$$

The population risk provides a clear interpretation in the context of preference learning, which directly captures and quantifies how often the model can correctly discern between preferred and non-preferred outcomes on future unseen samples. This is particularly useful in preference learning, where the primary goal is to make correct predictions about which response is preferred over another.

## 3.2 ANALYZE GPO UNDER NOISY FEEDBACK

Under the noise-free setting, Im & Li (2024a) analyzed generalization guarantees for models trained with preference optimization loss. However, human feedback can be inherently noisy. To capture a more practical setting, we aim to relax this strong assumption and instead analyze the generalization behavior of preference optimization under *noisy feedback*.

**$\epsilon$-noise preference data.** We consider a noisy preference dataset $\tilde{\mathcal{D}}_\epsilon = \{(x_i, \tilde{y}_{w,i}, \tilde{y}_{l,i})\}_{i=1}^N$, which flips the preference label with probability $\epsilon$ from $y_w \succ y_l$ to $y_l \succ y_w$ for samples in the noise-free oracle dataset $\mathcal{D} = \{(x_i, y_{w,i}, y_{l,i})\}_{i=1}^N$. Hence, $\epsilon$ captures the amount of noise in the dataset, where a larger $\epsilon$ means more severe noise contamination, and vice versa. This setup simulates the mistakes observed in both human-provided (Lindner & El-Assady, 2022) and heuristic-based preferences (Chen et al., 2024). Given the empirical noisy dataset $\tilde{\mathcal{D}}_\epsilon = \{(x_i, \tilde{y}_{w,i}, \tilde{y}_{l,i})\}_{i=1}^N$, we then fine-tune the LLM policy $\pi_\theta$ to minimize the GPO objective:

$$\mathbb{E}_{(x, \tilde{y}_w, \tilde{y}_l) \in \tilde{\mathcal{D}}_\epsilon} \left[ f\left( \beta \left( \log \frac{\pi_\theta(\tilde{y}_w | x)}{\pi_{\text{ref}}(\tilde{y}_w | x)} - \log \frac{\pi_\theta(\tilde{y}_l | x)}{\pi_{\text{ref}}(\tilde{y}_l | x)} \right) \right) \right], \tag{9}$$

where $\tilde{y}_w$ and $\tilde{y}_l$ are the noisy preferred and rejected labels for preference learning.

**Analyze GPO behavior under practical considerations.** A key focus of our paper is to provide a tractable analysis of GPO's generalization behavior, without divorcing from practical considerations. Our analytical framework is designed with practicality in mind. Besides taking noisy feedback into account, we consider the generalization of models after *finite gradient steps* when the loss is within a constant factor of its initial value. This scenario closely matches real-world practices, where large language models are often fine-tuned for a finite number of steps to avoid overfitting. For this reason, our analytical approach is different from classical generalization theory, which typically considers overparameterized models that achieve near-optimal loss (Allen-Zhu et al., 2019; Arora et al., 2019; Cao & Gu, 2020; Subramanian et al., 2022) or are independent of the training process (Arora et al., 2018; Lotfi et al., 2022; 2023).

Our theory revolves around analyzing how the reward margin changes over the course of training, which allows us to bound the generalization error after finite-step GPO updates. For an input prompt $x = (x^{(1)}, x^{(2)}, \ldots, x^{(T)})$ with length $T$, we denote the model output $f_\theta(x) = \text{softmax}(Wg(x))$, where $W$ is the unembedding matrix and $g(x)$ is the final hidden state. The feature backbone can be either fixed or tunable. For example, in recently popularized parameter-efficient fine-tuning paradigm, the feature backbone is often kept frozen to prevent overfitting (Hu et al., 2021; Houlsby et al., 2019), and in black-box fine-tuning scenarios where the backbone is not exposed to the end-user. In what follows, we first focus on a fixed encoder as a pragmatic approach to manage tractability while still extracting valuable insights into preference learning. Later we will also investigate whether our theoretical insights hold when performing full fine-tuning, where the feature map is allowed to change (Section 4).

We begin by stating a lemma on the gradient flow and reward dynamics.

**Lemma 3.1 (Gradient flow and reward dynamics).** *The dynamics of the reward margin for sample $j$ is given by*

$$\tau \dot{r_j}(t) = -\frac{1}{N} \sum_{i=1}^{N} \beta^2 f'(r_i(t)) (\tilde{\mathbf{y}}_{w,j} - \tilde{\mathbf{y}}_{l,j})^\top (\tilde{\mathbf{y}}_{w,i} - \tilde{\mathbf{y}}_{l,i}) \Sigma_{ij}, \tag{10}$$

*where $t$ is the time, $r_i$ is the shorthand notation for reward margin of sample $x_i$, $\Sigma$ is the sample covariance matrix with $\Sigma_{ij} = g(x_i)^\top g(x_j)$, and $\tau$ is inverse to the learning rate.*

*Proof.* To analyze the reward margin associated with each sample and its evolution during training, we begin by deriving the dynamics of the unembedding layer matrix $W$ under gradient flow:

$$\tau \dot{W} = -\frac{1}{N} \sum_{i=1}^{N} \beta f'(\beta (\tilde{\mathbf{y}}_{w,i} - \tilde{\mathbf{y}}_{l,i})^\top (W - W_0) g(x_i)) (\tilde{\mathbf{y}}_{w,i} - \tilde{\mathbf{y}}_{l,i}) g(x_i)^\top, \tag{11}$$

where $W_0$ is the initial weight in the reference policy $\pi_{\text{ref}}$. $\tau$ determines the rate of change, where a larger $\tau$ corresponds to a slower rate of change. $\tilde{\mathbf{y}}_{w,i}, \tilde{\mathbf{y}}_{l,i}$ are one hot vectors of the token, indicating either preferred or non-preferred. Let $\Delta W = W - W_0$, a constant offset from $W$, we have:

$$\tau \Delta \dot{W} = -\frac{1}{N} \sum_{i=1}^{N} \beta f'(\underbrace{\beta (\tilde{\mathbf{y}}_{w,i} - \tilde{\mathbf{y}}_{l,i})^\top \Delta W g(x_i)}_{\text{Reward margin for } x_i}) (\tilde{\mathbf{y}}_{w,i} - \tilde{\mathbf{y}}_{l,i}) g(x_i)^\top, \tag{12}$$

which contains the term of the reward margin. Since $\beta, \tilde{\mathbf{y}}_{w,j}, \tilde{\mathbf{y}}_{l,j}, x_j$ are fixed, we can consider the flow of the reward margin by multiplying $\beta (\tilde{\mathbf{y}}_{w,j} - \tilde{\mathbf{y}}_{l,j})^\top$ on the left and multiplying $g(x_j)$ on the right of $\tau \Delta \dot{W}$. This yields the dynamics of the reward margin. $\qquad \square$

**From training to test input reward dynamics.** We can extend this analysis beyond the training samples to any possible input. Consider a new triplet $(x^*, y_w^*, y_l^*)$ and let $r^*$ be its reward margin. While we do not train on this input, we can still follow its reward trajectory to derive the dynamics, which is given by

$$\tau \dot{r^*}(t) = -\frac{1}{N} \sum_{i=1}^{N} \beta^2 f'(r_i(t)) (\mathbf{y}_w^* - \mathbf{y}_l^*)^\top (\tilde{\mathbf{y}}_{w,i} - \tilde{\mathbf{y}}_{l,i}) g(x^*)^\top g(x_i). \tag{13}$$

By being able to follow the dynamics of the reward margins for any sample, we are able to reason about the shift in the decision boundary over the course of training, enabling us to establish a bound on the true population risk and quantify how the risk increases as noise is introduced.

## 3.3 GENERALIZATION GUARANTEE

We now characterize the preference distribution in order to provide a tractable analysis and bound the generalization error. Importantly, the features we model are designed to reflect the characteristics of the real-world transformer backbone, ensuring that our theoretical analysis remains grounded in the specific inductive biases and structures that are typical of such models. Specifically, we consider the sample embeddings are from a hyperspherical distribution with unit norm. This closely approximates the structure of embeddings observed after the RMSNorm layer in practical models such as LLaMA (Zhang & Sennrich, 2019; Touvron et al., 2023). In particular, we consider the von Mises-Fisher (vMF) distribution, a classical and important distribution in directional statistics (Mardia & Jupp, 2009), which is analogous to spherical Gaussian distributions for features with unit norms. The density function is given by $\rho(x; \mu, \kappa) = C_d(\kappa) e^{\kappa \mu^\top x}$, where $\mu$ represents the mean direction and $\kappa$ is the concentration parameter, and $C_d(\kappa)$ normalization constant dependent on the dimension $d$ and $\kappa$. We denote the distribution with mean direction $\mu$ and concentration parameter $\kappa$ as vMF$(\mu, \kappa)$. We also define a normalized concentration parameter $\gamma = \frac{2\kappa}{d}$. In **Appendix C**, we verify that embeddings from modern LLMs exhibit the key characteristics of the vMF distribution.

Under this characterization, we can now describe the data-generating process. First, we generate the set of positive samples $\mathcal{D}_+$, consisting of $N/2$ *i.i.d.* samples from vMF$(\mu_+, \kappa)$ and the set of negative samples $\mathcal{D}_-$, consisting of $N/2$ *i.i.d.* samples from vMF$(\mu_-, \kappa)$. Positive samples will have some preferred token $y_+$ and some rejected token $y_-$ while negative samples have the opposite

preferences. We define $2\theta$ to be the angle between $\mu_+$ and $\mu_-$. For each sample, we then generate an *i.i.d.* sample from a Bernoulli distribution with parameter $\epsilon$, flipping the sample's label if the outcome is 1. This results in our noisy dataset $\tilde{\mathcal{D}}_\epsilon = \tilde{\mathcal{D}}_+ \cup \tilde{\mathcal{D}}_-$. By using the reward dynamics as well as concentration results on the von Mises-Fisher distribution which we prove in **Appendix B**, we are able to bound the generalization error and capture how it changes with noise rate $\epsilon$.

**Theorem 3.1** (**Generalization guarantee under noisy feedback**). *Suppose we have a noisy dataset such that each sample has its labels flipped with probability $\epsilon$, with $0 \leq \epsilon \leq \frac{1}{2}$. Then, with probability at least $1 - \frac{2\mathcal{R}_0}{N - \epsilon N - \sqrt{\log N}} - \frac{2}{N^2}$, for $0 \leq \epsilon \leq \frac{1}{2}\left(1 - \frac{1}{\gamma} - \cos\frac{\theta}{3} - \frac{4\sqrt{\log N}}{N}\right)$ and $0 < t \leq \frac{\sin(\theta/3)\tau}{4\beta^2 D}$, the population risk of the model is bounded as*

$$\mathcal{R}(\mathcal{P}) \leq \frac{\mathcal{R}_0}{\left(1 - \sqrt{\mathcal{R}_0\gamma}\left(\epsilon + \frac{2\sqrt{\log N}}{N}\right)\right)^2}, \tag{14}$$

*where the clean risk bound under noise-free human feedback, $\mathcal{R}_0$, is given by*

$$\mathcal{R}_0 = \frac{4}{\gamma\left(1 - \frac{1}{\gamma} - \cos\frac{\theta}{3}\right)^2}. \tag{15}$$

**Theorem 3.2** (**Behavior of expected risk**). *Suppose we have a noisy dataset such that each sample has its label flipped with probability $\epsilon$. Then, for $0 \leq \epsilon \leq 1 - \frac{1}{\gamma} - \cos\frac{\theta}{3} - \frac{\sqrt{\log N}}{N}$ and $0 < t \leq \frac{\sin(\theta/3)\tau}{4\beta^2 D}$, the expected population risk of the model $\mathbb{E}_{\tilde{\mathcal{D}}_\epsilon}[\mathcal{R}(\mathcal{P})]$, averaged over the sampled noisy datasets $\tilde{\mathcal{D}}_\epsilon$, is bounded by*

$$\mathbb{E}_{\tilde{\mathcal{D}}_\epsilon}[\mathcal{R}(\mathcal{P})] \leq \frac{\mathcal{R}_0}{\left(1 - \sqrt{\mathcal{R}_0\gamma}\left(\epsilon + \frac{2\sqrt{\log N}}{N}\right)\right)^2} + \frac{2\mathcal{R}_0}{N - \epsilon N - \sqrt{\log N}} + \frac{2}{N^2}. \tag{16}$$

*Additionally, we have that for any $t$ and for any $\theta, \gamma$,*

$$\frac{d^2}{d\epsilon^2}\mathbb{E}_{\tilde{\mathcal{D}}_\epsilon}[\mathcal{R}(\mathcal{P})]\bigg|_{\epsilon=1/2} = 0 \tag{17}$$

**Theoretical insight on how the risk bound grows with $\epsilon$.** Unlike classical generalization theory, which typically analyzes model behavior at convergence, our theory leverages a finite-step analysis. This approach enables us to precisely reveal the impact of noisy labels in a fine-tuning setting. The key insight of the theorems is that given the bound on the risk for when there is no noise, $\mathcal{R}_0$, we can determine an upper bound on the rate at which the risk increases with $\epsilon$. In particular, as $\epsilon$ increases from 0, the bound increases as $1/(1 - \sqrt{\mathcal{R}_0\gamma}\epsilon)^2$ neglecting the finite-sample deviation for label flipping. With tighter bounds on the mean and variance of the cosine similarity between a sample and its corresponding mean, we can achieve tighter bounds on the noiseless risk and its rate of increase. As a result, we expect the risk in practice to be more closely modeled by

$$\frac{\mathbb{E}_{\mathcal{D}}[\mathcal{R}(\mathcal{P})]}{(1 - c\epsilon)^2} \tag{18}$$

for $\epsilon$ that is sufficiently away from $1/2$, where $\mathbb{E}_{\mathcal{D}}[\mathcal{R}(\mathcal{P})]$ is the risk of the model averaged over sampled noiseless datasets, and $c$ is a parameter that depends on the data distribution and training configuration. For $\epsilon$ near $1/2$, based upon the theorems, we expect an inflection point in the expected risk at $\epsilon = 1/2$, and therefore, we can expect the test accuracy, as $\epsilon$ approaches $1/2$, to decrease at an approximately linear rate. We empirically observe that this theory-based model of the risk growing as $\frac{1}{(1-c\epsilon)^2}$ and transitioning to linearity near $\epsilon = 0.5$ closely describes the test accuracy on real-world datasets in Section 4. This suggests that building upon our theoretical results can lead to a close match between theory and practice.

Additionally, we can understand how the risk bound varies with parameters of the data distribution. As both $\theta$ (distance between the mean of the two distributions) and $\gamma$ (concentration within each distribution) increase, $\mathcal{R}_0$ decreases. In particular, $\mathcal{R}_0$ is approximately inversely proportional to $\gamma$, and $\mathcal{R}_0$ is inversely proportional to $1 - \frac{1}{\gamma} - \cos\frac{\theta}{3}$. Moreover, increasing $\gamma$ and $\theta$ leads to an

increase in $\sqrt{\mathcal{R}_0 \gamma}$, which governs the rate at which the risk bound grows with $\epsilon$. A larger $\sqrt{\mathcal{R}_0 \gamma}$ results in a slower increase in risk, meaning that greater $\gamma$ and $\theta$ contribute to a slower rise in risk as $\epsilon$ approaches 0. In summary, less similarity between positive and negative examples, along with more concentrated distributions, allows for a tighter bound on population risk and less sensitivity to noise for smaller $\epsilon$.

**Derivation overivew.** We provide a high-level summary of the derivation of the risk bound. In the noiseless case, the initial direction of the GPO update will always correspond to a sample estimate of the difference between the means of the positive and negative example distributions. This estimate becomes more robust as the number of samples increases, the distance between means increases, and as the distributions become more concentrated, and as a result, the risk increases at a slower rate with respect to the noise rate. Furthermore, by reasoning about the reward margin for any sample, including those outside the training distribution (*cf* Equation 13), we can control how the decision boundary shifts by the end of training. We then analyze which samples would be classified correctly, accounting for estimation error in the mean difference due to noise and finite samples, as well as the boundary shift from training. Using tail bounds, we can provide a guarantee for the risk when $\epsilon$ is small. In order to determine how the expectation of the risk behaves as $\epsilon$ approaches $\frac{1}{2}$, we use the symmetry of the expected risk over $1/2$ to determine that there is an inflection point at which the risk approaches a linear rate.

In Section 4, we empirically validate our theoretical bound by training on contemporary LLMs such as LLaMa (Touvron et al., 2023), where we observe the predicted behavior. Moreover, we extend this analysis to full fine-tuning scenarios in large language models, demonstrating that the insight holds broadly and offering practical guidance.

> **Key takeaways of Section 3**
>
> 1. Our theory suggests that the expected risk can be modeled as $\frac{\mathbb{E}_{\mathcal{D}}[\mathcal{R}(\mathcal{P})]}{(1-c\epsilon)^2}$, which is a function of the noiseless risk and $\epsilon$ for $\epsilon$ sufficiently below $1/2$.
>
> 2. As $\epsilon$ approaches $1/2$, the expected risk decreases approximately linearly as it approaches an inflection point.
>
> 3. Stronger concentration, more samples, and contrasting directions for positive and negative samples allow for tighter bounds and slower degradation in accuracy as the noise rate increases.

## 4 CONNECTING THEORY TO PRACTICE

To understand how our theory guides practical LLM training, we verify the generalization behavior of preference optimization when updating last-layer parameters and updating all model parameters. In particular, Section 4.1 focuses on experiments conducted within a controlled setting, which allows us to systematically verify the impact of noise rate and distributional properties on model performance. In Section 4.2, we extend our investigation to a real-world dataset, to validate the practical applicability of our findings in a more complex and realistic setting. Section 4.3 verifies that our theoretical insights indeed hold on other preference optimization losses in the GPO family.

### 4.1 VERIFICATION OF BOUND IN A CONTROLLED SETTING

**Experimental setup.** We first validate the risk bound in a controlled setting where we can flexibly parameterize the data distribution. We consider data points with dimension $d = 512$, sampled from vMF distribution, with the mean vectors for the positive and negative samples separated by an angle of $2\theta$. To study the effects of $\gamma$ and $\theta$, we vary the concentration parameter $\gamma$ over values $1/16$, $1/8$, and $1/4$ while keeping $\theta$ fixed at $\pi/3$, and vary $\theta$ over $\pi/3$, $2\pi/3$, and $\pi$ with $\gamma$ fixed at $1/8$. We sample 1000 data points each from the positive and negative distributions, with $\epsilon$ ranging from 0 to $1/2$ in increments of 0.025. The model, which has two outputs corresponding to positive and negative samples, is trained with DPO loss for 10 epochs using gradient descent. For each configuration, we perform 20 trials and plot the average test accuracy as a function of $\epsilon$. Additionally, we fit the theoretical model from Equation 18 to the data for $\epsilon = 0$ to 0.35, assuming that the true noiseless risk is at most 1% from the observed average test error. We present the results in Figure 1.

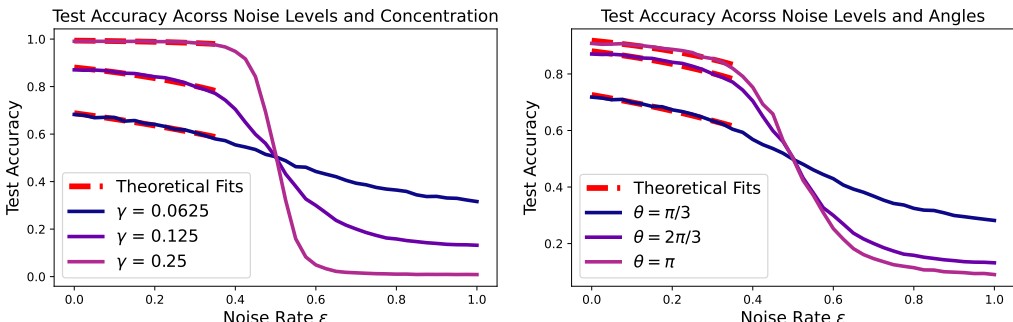

Figure 1: Empirical validation in a controlled setting with (left) concentration parameter $\gamma$ varying over $1/16, 1/8, 1/4$ and with (right) $\theta$ varying over $\pi/3, 2\pi/3, \pi$. In both plots, we vary the noise rate $\epsilon$ on the $x$-axis from 0 to 1 with increments of 0.025. All curves are averaged over 20 runs.

**Impact of noise rate $\epsilon$.** In Figure 1, we plot how the test accuracy of model changes with increasing noise rate $\epsilon$. The figure aligns with our theoretical analysis of how the generalization error in preference learning increases as the noise rate rises. In particular, we can observe that the theoretical fit closely follows the empirical accuracy observed, validating the theory that the growth in the expected risk for noisy datasets is well approximated by $\frac{1}{(1-c\epsilon)^2}$ for $\epsilon$ smaller than 0.5. Additionally, we observe an inflection point around $\epsilon = 0.5$, where the test accuracy begins to decrease approximately linearly.

**Impact of distribution parameters.** We can observe in Figure 1a that as $\gamma$ increases and in Figure 1b that as $\theta$ increases, the noiseless test accuracy is generally higher (when the noise rate is under 0.5). Moreover, when $\gamma$ or $\theta$ increases, the test accuracy decreases at a slower rate when $\epsilon$ is closer to 0. These empirical results match the relationship between the distributional parameters and the risk discussed in detail in the theoretical insights.

### 4.2 VERIFICATION ON REAL-WORLD DATASET

**Experimental setup.** To further verify our theory on real-world dataset, we use HH-RLHF (Bai et al., 2022a), a dataset consisting of human preferences about helpfulness and harmlessness with 161k training samples and 8.55k test samples[1]. We format each sample to be in the form of a prompt and two responses, with one being preferred over the other, and we exclude samples that did not fit this format resulting in 160k training samples and 8.53k test samples. We perform **full fine-tuning** on the Llama-2-7B model (Touvron et al., 2023) using the DPO loss. This allows us to validate our theory, updating all parameters, and thus provides more complete empirical validation. We train with noise rates ranging from $\epsilon = 0$ to $\epsilon = 0.5$ with 0.05 increments, and measure the test performance for each setting. Specifically, we perform SFT for 1 epoch on the preferred response to each prompt in the noisy training set, where each training sample had its labels flipped with probability $\epsilon$. We then perform DPO for 1 epoch on the same noisy dataset. As in Section 4.1, we plot the best fit of our theory-based model in Equation 18, assuming the true noiseless risk deviates from the observed average test error by no more than 1%. We provide the complete training hyperparameters in Appendix A.

**Our theoretical implication holds on real-world dataset with full fine-tuning.** For the HH-RLHF dataset, we can see in Figure 2 that the accuracy decreases at a near constant rate. This is due to the fact that about 30% of the labels are already noisy (Wang et al., 2024), and as the

---

[1] https://huggingface.co/datasets/Anthropic/hh-rlhf

true range of the noise rate we consider is approximately ranging from 0.3 to 0.5[2], *we expect the decline in accuracy to already be transitioning towards linearity according to our Theorem 3.2.* Our theory-based model maintains a close fit to the observed test accuracies, further validating our theoretical framework. While a similar trend is observed in Gao et al. (2024b), their work is purely empirical, lacking the rigorous theoretical foundation that we provide. Our theoretical contribution offers a precise explanation of the behavior of test accuracy as $\epsilon$ increases, as well as the transition to a linear decline, which aligns with the empirical results. Overall, the close match between our theoretical analysis and empirical observation highlights the strength and applicability of

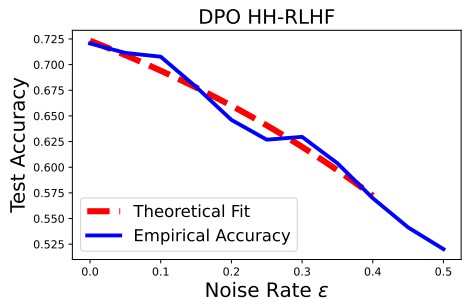

Figure 2: Test accuracy for HH-RLHF across varying noise rates $\epsilon$.

our theoretical framework in modeling the effects of noise on preference optimization.

### 4.3 VERIFICATION ON DIFFERENT LOSSES IN GPO FAMILY

**Our theory holds on alternative loss in GPO family.** We extend our experiments to the IPO objective (Azar et al., 2023) to confirm that our theoretical insights are not specific to DPO but hold for other objectives in the GPO family. We keep the experimental setting the same as in Section 4.1 and provide the results in Figure 3. We can see that the theory-based model matches the empirical average test accuracy well where it starts to transition to a linear decrease. Moreover, in Figure 3, we observe the expected inverse relationship between the parameters $\gamma$ and $\theta$ and the risk for IPO, further validating the applicability of our analysis. This consistency highlights the broad applicability of our theoretical framework to different preference optimization objectives.

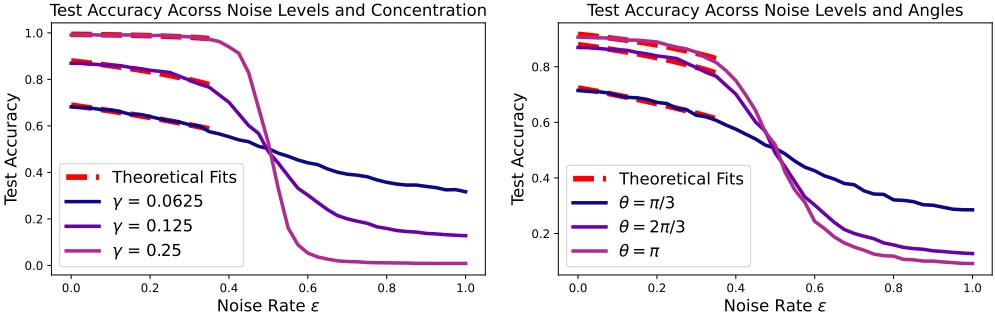

Figure 3: Empirical validation using IPO loss in the controlled setting. **Left**: concentration parameter $\gamma$ varies over $1/16, 1/8, 1/4$. **Right**: $\theta$ varies over $\pi/3, 2\pi/3, \pi$. In both plots, we vary the noise rate $\epsilon$ on the $x$-axis from 0 to 1 with increments of 0.025. All curves are averaged over 20 runs.

## 5 RELATED WORKS

**Alignment of LLMs.** A key aspect of training and deploying large language models is ensuring the models behave in safe and helpful ways (Ji et al., 2023; Casper et al., 2023; Hendrycks et al., 2021; Leike et al., 2018). This is an important problem due to the potential harms that can arise in large models (Park et al., 2023; Carroll et al., 2023; Perez et al., 2022; Sharma et al., 2023; Bang et al., 2023; Hubinger et al., 2019; Berglund et al., 2023; Ngo et al., 2022; Shevlane et al., 2023; Shah et al., 2022; Pan et al., 2022). A wide range of methods have been developed that utilize human feedback or human preference data to train models to avoid harmful responses and elicit

---

[2]With 30% initial noise, flipping the preference label with $\epsilon = 0.5$ results in 15% of the incorrect labels becoming correct. Meanwhile, from the 70% of initially correct labels, 35% remain correct. Overall, this brings the total noise level to 50%.

safer or more helpful responses (Christiano et al., 2017; Ziegler et al., 2019; Stiennon et al., 2020; Lee et al., 2021; Ouyang et al., 2022; Bai et al., 2022a; Nakano et al., 2022; Glaese et al., 2022; Snell et al., 2023; Yuan et al., 2023; Song et al., 2023; Dong et al., 2023; Bai et al., 2022b; Lee et al., 2023; Munos et al., 2023; Hejna et al., 2023; Dai et al., 2023; Khanov et al., 2024). Particularly, the Reinforcement Learning from Human Feedback (RLHF) framework has proven effective in aligning large pre-trained language models (Christiano et al., 2017; Ziegler et al., 2019; Ouyang et al., 2022; Bai et al., 2022a). However, given its computational inefficiency, recent shifts in focus favor closed-form losses that directly utilize offline preferences, like Direct Preference Optimization (DPO) (Rafailov et al., 2023) and related methodologies (Azar et al., 2023; Pal et al., 2024; Liu et al., 2024b; Ethayarajh et al., 2024a; Xiong et al., 2023; Tang et al., 2024; Meng et al., 2024; Ethayarajh et al., 2024b; Zeng et al., 2024; Calandriello et al., 2024; Muldrew et al., 2024; Ray Chowdhury et al., 2024; Liu et al., 2024a; Gao et al., 2024a; Yang et al., 2024; Chakraborty et al., 2024). Despite the empirical success and wide adoption in real-world systems (OpenAI, 2023; Anthropic, 2023; Touvron et al., 2023), fewer works provide theoretical underpinnings (Azar et al., 2023; Rafailov et al., 2024; Im & Li, 2024b; Tang et al., 2024; Ray Chowdhury et al., 2024; Tajwar et al., 2024; Xu et al., 2024; Nika et al., 2024; Xiong et al., 2024). In this work, we make an initial attempt to theoretically analyze the generalization behavior of preference optimization under noisy feedback, making our results particularly relevant for the development and deployment of robust LLM systems.

**Robustness of preference optimization.** Ensuring that a model can generalize when trained with noisy labels is crucial for building robust and reliable systems (Song et al., 2022). This problem has led to a wide range of works Song et al. (2022) developing various methods that improve model generalization in the presence of noise with many of the works presenting theoretical guarantees of robustness (Natarajan et al., 2013; Zhang & Sabuncu, 2018; Li et al., 2020) for modified loss functions or for early stopping. In the context of preference learning, increased noise levels have been shown to degrade performance, especially when considering loss minimizers (Gao et al., 2024b; Fisch et al., 2024; Liang et al., 2024). This has led to the development of methods such as ROPO (Liang et al., 2024), cDPO (Mitchell, 2023), and rDPO (Ray Chowdhury et al., 2024) which introduce modifications to the DPO objective and its gradients. Fisch et al. (2024) considers a pessimistic distillation loss to learn rewards robustly. These approaches have proven effective in enhancing the robustness of preference optimization. Complementing these efforts, our study provides a rigorous generalization analysis of finite-step preference optimization under noisy feedback. Our theory, grounded in reward dynamics, offers new insights on how the population risk grows with the noise rate for offline preference learning in a finite-step training setting.

## 6 CONCLUSION

Our work theoretically analyzes the generalization behavior of preference learning in the presence of noisy labels through a dynamics-based approach based on a general class of objectives, including methods such as DPO, IPO, SLiC, etc., which implicitly learn a reward model. Key to our framework, we analyze the reward margin associated with each training sample and its trajectory throughout the training process, enabling us to effectively bound the generalization error. Through rigorous analysis and novel bounds, we establish a generalization guarantee that depends on the noise rate and provide a model based upon the theoretical guarantee that closely describes how test accuracy is impacted by noise on real-world datasets. Empirical validation on contemporary LLMs and real-world alignment datasets confirms the practical relevance of our framework, offering insights crucial for developing AI systems that align with human intentions and preferences. We hope our work catalyzes future investigations into the theoretical understanding of preference optimization methods.

## LIMITATION

While our work provides new theoretical insights into preference optimization under noisy feedback, it does have its constraints. Notably, our framework is limited to offline settings, which assumes that the feedback is collected apriori. Analyzing generalization behavior in online RL settings remains a significant challenge. This limitation underscores the necessity for future research to further explore the theoretical understanding of preference optimization.

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

## A    ADDITIONAL EXPERIMENTAL DETAILS

We provide the hyperparameters used for experiments.

Table 1: Summary of training hyperparameters for supervised fine-tuning and DPO for Llama-2-7B for HH-RLHF.

|  | Parameters | Value |
|---|---|---|
| Supervised fine-tuning | Number of epochs | 1 |
|  | Optimizer | AdamW |
|  | Learning rate | $10^{-5}$ |
|  | Batch size | 256 |
|  | Gradient accumulation steps | 1 |
|  | Maximum sequence length | 512 |
|  | DeepSpeed Zero stage | 3 |
|  | Weight decay | 0 |
| DPO/IPO | Number of epochs | 1 |
|  | Optimizer | AdamW |
|  | Learning rate | $10^{-5}$ |
|  | $\beta$ | 0.1 |
|  | Batch size | 256 |
|  | Gradient accumulation steps | 1 |
|  | Maximum sequence length | 512 |
|  | DeepSpeed Zero stage | 3 |
|  | Max prompt length | 256 |
|  | Max target length | 256 |
|  | Weight decay | 0 |

# B PROOF OF THEOREM 3.1

We start by proving concentration results on the von Mises-Fisher distribution.

**Lemma B.1** (von Mises-Fisher Tail Bound). *Given an i.i.d. sample $x$ from the von Mises Fisher Distribution with mean $\mu$ and concentration $\kappa = \gamma\left(\frac{d}{2}\right)$ for $\gamma \geq 4$, with probability at least $1 - \frac{1}{\alpha^2}$,*

$$x^\top \mu \geq \frac{\sqrt{1+\gamma^2}-1}{\gamma} - \alpha\sqrt{\frac{4}{\gamma}} \tag{19}$$

**Proof.** We first start by determining a lower bound for the expected value of $x^\top \mu$. This is given by

$$\frac{I_{d/2}(\kappa)}{I_{d/2-1}(\kappa)} \tag{20}$$

where $I_{d/2}$ is the Modified Bessel function of the first kind. Then, by Laforgia & Natalini (2010), Theorem 1.1, we have that

$$\frac{I_{d/2}(\kappa)}{I_{d/2-1}(\kappa)} > \frac{-\frac{d}{2} + \sqrt{\left(\frac{d}{2}\right)^2 + \kappa^2}}{\kappa} \tag{21}$$

Then, defining $\gamma$ through $\kappa = \frac{\gamma d}{2}$, we have

$$\mathbb{E}[x^\top \mu] > \frac{\sqrt{1+\gamma^2}-1}{\gamma} \tag{22}$$

Now, we will upper bound the variance of $x^\top \mu$. In order to do so, we need an upper bound on $\mathbb{E}[(x^\top \mu)^2]$. Notice that this expectation is equal to

$$C_d(\kappa) \int_{\mathbb{S}^{d-1}} e^{\kappa x^\top \mu} (x^\top \mu)^2 dx \tag{23}$$

where $C_d(\kappa)$ is the normalizing constant and that

$$C_d(\kappa) \int_{\mathbb{S}^{d-1}} e^{\kappa x^\top \mu} (x^\top \mu)^2 dx = C_d(\kappa) \frac{d^2}{d\kappa^2} \int_{\mathbb{S}^{d-1}} e^{\kappa x^\top \mu} dx \tag{24}$$

Then, we have that

$$C_d(\kappa) \frac{d^2}{d\kappa^2} \int_{\mathbb{S}^{d-1}} e^{\kappa x^\top \mu} dx = \frac{\kappa^{d/2-1}}{(2\pi)^{d/2} I_{d/2-1}(\kappa)} \frac{d^2}{d\kappa^2} \left( \frac{(2\pi)^{d/2} I_{d/2-1}(\kappa)}{\kappa^{d/2-1}} \right) \tag{25}$$

and this can be simplified as

$$\frac{\kappa^{d/2-1}}{I_{d/2-1}(\kappa)} \frac{d^2}{d\kappa^2} \left( \frac{I_{d/2-1}(\kappa)}{\kappa^{d/2-1}} \right) = \frac{\kappa^{d/2-1}}{I_{d/2-1}(\kappa)} \frac{d}{d\kappa} \left( \frac{I'_{d/2-1}(\kappa)}{\kappa^{d/2-1}} - \frac{(d/2-1) I_{d/2-1}(\kappa)}{\kappa^{d/2}} \right) \tag{26}$$

and further as

$$\frac{\kappa^{d/2-1}}{I_{d/2-1}(\kappa)} \frac{d}{d\kappa} \left( \frac{I'_{d/2-1}(\kappa)}{\kappa^{d/2-1}} - \frac{(d/2-1) I_{d/2-1}(\kappa)}{\kappa^{d/2}} \right)$$

$$= \frac{\kappa^{d/2-1}}{I_{d/2-1}(\kappa)} \left( \frac{I''_{d/2-1}(\kappa)}{\kappa^{d/2-1}} - \frac{(d-2) I'_{d/2-1}(\kappa)}{\kappa^{d/2}} - \frac{(d^2/4 - d/2) I_{d/2-1}(\kappa)}{\kappa^{d/2+1}} \right)$$

$$= \left( \frac{I''_{d/2-1}(\kappa)}{I_{d/2-1}(\kappa)} - \frac{(d-2) I'_{d/2-1}(\kappa)}{\kappa I_{d/2-1}(\kappa)} - \frac{(d^2/4 - d/2)}{\kappa^2} \right) \tag{27}$$

Then, using the identity from Wolfram (2001), we have that

$$\left( \frac{I''_{d/2-1}(\kappa)}{I_{d/2-1}(\kappa)} - \frac{(d-2) I'_{d/2-1}(\kappa)}{\kappa I_{d/2-1}(\kappa)} - \frac{(d^2/4 - d/2)}{\kappa^2} \right)$$

$$\leq \frac{I''_{d/2-1}(\kappa)}{I_{d/2-1}(\kappa)} - \frac{I'_{d/2-1}(\kappa)}{\gamma I_{d/2-1}(\kappa)} - \frac{1}{2\gamma^2}$$

$$= \frac{1}{2} + \frac{I_{d/2-3}(\kappa) + I_{d/2+1}}{4 I_{d/2-1}(\kappa)} - \frac{I_{d/2-2}(\kappa) + I_{d/2}(\kappa)}{2\gamma I_{d/2-1}(\kappa)} - \frac{1}{2\gamma^2} \tag{28}$$

Then, by Theorem 1.1 from Laforgia & Natalini (2010), and the fact that $\frac{1}{\sqrt{x^2+\kappa^2}-x}$ is an increasing function for $x > 0$, we have that

$$\frac{1}{2} + \frac{I_{d/2-3}(\kappa) + I_{d/2+1}}{4I_{d/2-1}(\kappa)} - \frac{I_{d/2-2}(\kappa) + I_{d/2}(\kappa)}{2\gamma I_{d/2-1}(\kappa)} - \frac{1}{2\gamma^2}$$

$$\leq \frac{3}{4} + \frac{\gamma^2}{4(\sqrt{1+\gamma^2}-1)^2} - \frac{\sqrt{1+\gamma^2}-1}{2\gamma^2} - \frac{1}{2\gamma} - \frac{1}{2\gamma^2} \quad (29)$$

Then, the variance of $x^\top\mu$ is upper bounded by

$$\frac{3}{4} + \frac{\gamma^2}{4(\sqrt{1+\gamma^2}-1)^2} - \frac{\sqrt{1+\gamma^2}-1}{2\gamma^2} - \frac{1}{2\gamma} - \frac{1}{2\gamma^2} - \frac{(\sqrt{1+\gamma^2}-1)^2}{\gamma^2} \quad (30)$$

Given that $\gamma \geq 4$, we have that

$$\frac{\gamma^2}{4(\sqrt{1+\gamma^2}-1)^2} \leq \frac{1}{4} + \frac{3}{\gamma} \quad (31)$$

resulting in an upper bound of

$$1 + \frac{5}{2\gamma} - \frac{\sqrt{1+\gamma^2}-1}{2\gamma^2} - \frac{1}{2\gamma^2} - \frac{(\sqrt{1+\gamma^2}-1)^2}{\gamma^2} \quad (32)$$

and as

$$\frac{\sqrt{1+\gamma^2}-1}{\gamma} \geq 1 - \frac{1}{\gamma} \quad (33)$$

we have an upper bound of

$$1 + \frac{2}{\gamma} - \left(1 - \frac{1}{\gamma}\right)^2 \quad (34)$$

and as

$$\left(1 - \frac{1}{\gamma}\right)^2 \geq 1 - \frac{2}{\gamma} \quad (35)$$

we have that the variance is upper bounded by

$$\frac{4}{\gamma} \quad (36)$$

Then, applying Chebyshev's inequality with the upper bound on the variance gives the desired result. $\qquad\square$

**Lemma B.2** (von Mises-Fisher Mean Concentration). *Given $N$ i.i.d. samples $x_1, x_2, \ldots, x_N$ from the von Mises Fisher Distribution with mean $\mu$ and concentration $\kappa = \gamma\left(\frac{d}{2}\right)$ for $\gamma \geq 4$, with probability at least $1 - \frac{1}{\alpha^2}$,*

$$\frac{1}{N}\sum_{i=1}^{N} x_i^\top\mu \geq \frac{\sqrt{1+\gamma^2}-1}{\gamma} - \alpha\sqrt{\frac{4}{N\gamma}} \quad (37)$$

**Proof.** Since $x_1, x_2, \ldots, x_N$ area i.i.d. it follows that the variance of dot product of $\mu$ and the mean of the $N$ samples is $N$ times smaller than the variance of $x_i^\top\mu$. Then, applying the upper bound on the variance from Lemma B.1 as well as Chebyshev's inequality, we have the desired result. $\qquad\square$

**Lemma B.3** (Training Boundary Shift). *For $0 < t \leq \frac{\delta\tau}{4\beta^2 D}$, the angle between the boundary at time $t$ and the initial boundary is at most $\arcsin\delta$ for $0 < \delta < 1$.*

**Proof.** We start with the case for $f$ with $f'(0) < 0$ and $|f''(x)| \leq D$ for $x \geq 0$. As the weights follow the following dynamics,

$$\tau \Delta \dot{W} = -\frac{1}{N} \sum_{i=1}^{N} \beta f'(\underbrace{\beta(\tilde{\mathbf{y}}_{w,i} - \tilde{\mathbf{y}}_{l,i})^\top \Delta W g(x_i)}_{\text{Reward margin for } x_i})(\tilde{\mathbf{y}}_{w,i} - \tilde{\mathbf{y}}_{l,i})g(x_i)^\top, \tag{38}$$

we can say that the initial direction that the weights are along is

$$-\frac{1}{N} \sum_{i=1}^{N} \beta f'(0)(\tilde{\mathbf{y}}_{w,i} - \tilde{\mathbf{y}}_{l,i})g(x_i)^\top \tag{39}$$

which we will define as $W_{0+}$. We aim to control the angle between the initial boundary and the boundary at time $t$. To do so, consider any sample $x^*$ with corresponding reward $r^*$. Then, we know that at $t = 0$,

$$\tau \dot{r^*}(0) = \beta(\mathbf{y}_w^* - \mathbf{y}_l^*)^\top W_{0+} g(x^*). \tag{40}$$

Now, let $B_0 = (\mathbf{y}_w^* - \mathbf{y}_l^*)^\top W_{0+}$, and suppose the cosine similarity between $B_0, g(x^*)$ is greater than or equal to $\delta$. Then,

$$\tau \dot{r^*}(0) \geq \beta \|B_0\| \delta \tag{41}$$

Now, we will determine a lower bound, $t_s$, for $t^*$ which is defined as the first time $|\tau \dot{r^*}(t) - \tau \dot{r^*}(0)| = \beta \|B_0\| \delta$, and the lower bound should hold for any sample that satisfies the equation above as this will guarantee that the boundary shifts by at most an angle of $\arcsin \delta$ at time $t_s$. First, we bound the magnitude of the second time derivative of the reward which has the form

$$\tau \ddot{r^*}(t) = -\frac{1}{N} \sum_{i=1}^{N} \beta^2 f''(r_i) \dot{r^*}(t)(\mathbf{y}_w^* - \mathbf{y}_l^*)^\top (\tilde{\mathbf{y}}_{w,i} - \tilde{\mathbf{y}}_{l,i})g(x^*)^\top g(x_i) \tag{42}$$

Since we consider $f$ with second derivative with magnitude bounded by $D$ and unit norm embeddings,

$$|\ddot{r^*}(t)| \leq \frac{2\beta^2 D}{\tau} |\dot{r^*}(t)| \tag{43}$$

Since we consider time up to $t_s$, we know that $|\dot{r^*}(t)| \leq 2\beta \|B_0\|$. Then, it follows that

$$|\ddot{r^*}(t)| \leq \frac{4\beta^3 D \|B_0\|}{\tau^2} \tag{44}$$

Then, we have that

$$|\dot{r^*}(t) - \dot{r^*}(0)| \leq \frac{4\beta^3 D \|B_0\| t}{\tau^2} \tag{45}$$

Then, as we need $|\tau \dot{r^*}(t) - \tau \dot{r^*}(0)| \leq \beta \|B_0\| \delta$, we can lower bound $t_s$ by

$$\frac{\delta \tau}{4\beta^2 D} \tag{46}$$

Then, it follows that for $0 < t \leq \frac{\delta \tau}{4\beta^2 D}$, the angle between the boundary at time $t$ and the initial boundary is at most $\arcsin \delta$.

In the case of SLiC, since $f'(x) = 1$ for $0 \leq x < 1$, we can ensure that the boundary actually stays the same as initialization as long as we stop before any reward is greater than or equal to 1. We can ensure this by bounding $|\dot{r^*}(t)|$ for any sample $r^*$. Based on the fact that $f'(x) = 1$ for $0 \leq x < 1$ and that we will only have rewards in this range, we have that

$$|\dot{r^*}(t)| \leq \frac{2\beta}{\tau} \tag{47}$$

Then, since $\delta < 1$, at any time $0 < t \leq \frac{\delta \tau}{2\beta}$, $r^*(t) \leq \delta$ for any $r^*$, and since $\delta < 1$, we have that the boundary will not shift from the initial direction during this range of time. Then, since we set $D = \frac{1}{2\beta}$ for SLiC, this completes the proof. $\qquad\square$

**Lemma B.4** (Generalization Error with Clean Samples). *Suppose we have a dataset of $N$ samples with half being positive and half being negative. Suppose that the cosine similarity between $\mu_+$ and $\mu_-$ is less than or equal to $\cos(2\theta)$ with $0 < \theta \leq \frac{\pi}{2}$. Then, with probability at least $1 - \frac{2\mathcal{R}_0}{N}$, we have that for $0 < t \leq \frac{\sin(\theta/3)\tau}{4\beta^2 D}$, the population risk of the model is bounded as*

$$\mathcal{R}(\mathcal{P}) \leq \mathcal{R}_0 \tag{48}$$

*where*

$$\mathcal{R}_0 = \frac{8}{\gamma \left(1 - \frac{1}{\gamma} - \cos\frac{\theta}{3}\right)^2} \tag{49}$$

**Proof.** By Lemma B.2, we have that with probability at least $1 - \frac{2\mathcal{R}_0}{N}$

$$\frac{2}{N} \sum_{i=1}^{N/2} x_i^{(+)\top} \mu_+ \geq \cos\frac{\theta}{3} \tag{50}$$

Then, as empirical mean $\frac{2}{N} \sum_{i=1}^{N/2} x_i^{(+)}$ has at most unit norm, we know that it is within an angle of $\theta/3$ from $\mu_+$. Similarly, by Lemma B.2, we have that with probability at least $1 - \frac{2\delta}{N}$

$$\frac{2}{N} \sum_{i=1}^{N/2} x_i^{(-)\top} \mu_- \geq \cos\frac{\theta}{3} \tag{51}$$

Then, as empirical mean $\frac{2}{N} \sum_{i=1}^{N/2} x_i^{(-)}$ has at most unit norm, we know that it is within an angle of $\theta/3$ from $\mu_-$. Then, it follows that

$$\frac{1}{N} \left( \sum_{i=1}^{N/2} x_i^{(+)} - \sum_{i=1}^{N/2} x_i^{(-)} \right) \tag{52}$$

is within an angle of $\theta/3$ from $\mu_+ - \mu_-$. Therefore, the resulting initial boundary direction is within an angle of $\theta/3$ from that corresponding to $\mu_+ - \mu_-$. By Lemma B.3, we know that for $0 < t \leq \frac{\sin(\theta/3)\tau}{4\beta^2 D}$, the boundary at time $t$ is within an angle of $\theta/3$ from the initial boundary. Then, as $\mu_+ - \mu_-$ is $\theta$ away from each of $\mu_+, \mu_-$, we know that any sample within an angle of $\theta/3$ from the corresponding mean will be classified correctly. For a new sample, by Lemma B.1, this occurs with probability at least $1 - \delta$, and therefore the risk is upper bounded by $\mathcal{R}_0$ or

$$\mathcal{R}(\mathcal{P}) \leq \mathcal{R}_0 \tag{53}$$

$\square$

**Lemma B.5** (Concentration for Bernoulli). *Suppose we have $N$ i.i.d. $Ber(\epsilon)$ random variables, $z_1, z_2, \ldots, z_N$. Then, with probability at least $1 - \frac{2}{N^2}$*

$$\left| \frac{1}{N} \sum_{i=1}^{N} -\epsilon \right| \leq \frac{\sqrt{\log N}}{N} \tag{54}$$

**Proof.** The result follows directly from Hoeffding's inequality. $\square$

**Lemma B.6** (Directional Perturbation from Noise). *Suppose we have a noisy dataset, with each sample having its labels flipped with probability $\epsilon$, with $0 \leq \epsilon \leq \frac{1}{2}$. Let $\tilde{x}_1^{(+)}, \tilde{x}_2^{(+)}, \ldots, \tilde{x}_{N_+}^{(+)}$ be the resulting set of samples that have labels corresponding to positive examples, and let $\tilde{x}_1^{(-)}, \tilde{x}_2^{(-)}, \ldots, \tilde{x}_{N_-}^{(-)}$ be the set of negative examples. Then, with probability at least $1 - \frac{2}{\alpha^2} - \frac{2}{N^2}$ we have that*

$$\frac{1}{N_+} \sum_{i=1}^{N_+} \mu_+^\top \tilde{x}_i^{(+)} \geq 1 - 2\epsilon - \frac{4\sqrt{\log N}}{N} - \frac{1}{\gamma} - \alpha \sqrt{\frac{8}{(N - \epsilon N - \sqrt{\log N})\gamma}} \tag{55}$$

$$\frac{1}{N_-} \sum_{i=1}^{N_-} \mu_-^\top \tilde{x}_i^{(-)} \geq 1 - 2\epsilon - \frac{4\sqrt{\log N}}{N} - \frac{1}{\gamma} - \alpha \sqrt{\frac{8}{(N - \epsilon N - \sqrt{\log N})\gamma}} \tag{56}$$

**Proof.** Let $N_{++}$ be the number of samples that were originally labeled positive and remained positive after the label flipping, and let $N_{-+} = \frac{N}{2} - N_{++}$ be the number of samples that were originally labeled positive and had their labels flipped. Similarly, let $N_{--}$ be the number of samples that were originally labeled negative and remained negative after the label flipping, and let $N_{+-} = \frac{N}{2} - N_{--}$ be the number of samples that were originally labeled negative and had their labels flipped. We will arrange the samples such that those that did not have their labels flipped correspond to the first $N_{++}$ or $N_{--}$ indices. Then,

$$\frac{1}{N_+} \sum_{i=1}^{N_+} \mu_+^\top \tilde{x}_i^{(+)} = \frac{1}{N_+} \left( \sum_{i=1}^{N_{++}} \mu_+^\top \tilde{x}_i^{(+)} + \sum_{i=N_{++}+1}^{N/2} \mu_+^\top \tilde{x}_i^{(+)} \right) \tag{57}$$

$$\frac{1}{N_-} \sum_{i=1}^{N_-} \mu_-^\top \tilde{x}_i^{(-)} = \frac{1}{N_-} \left( \sum_{i=1}^{N_{--}} \mu_-^\top \tilde{x}_i^{(-)} + \sum_{i=N_{--}+1}^{N/2} \mu_-^\top \tilde{x}_i^{(-)} \right) \tag{58}$$

Then, as $\mu_+, \mu_-$ and all sample embeddings have unit norm, we have

$$\frac{1}{N_+} \sum_{i=1}^{N_+} \mu_+^\top \tilde{x}_i^{(+)} = \frac{1}{N_+} \sum_{i=1}^{N_{++}} \mu_+^\top \tilde{x}_i^{(+)} - \frac{N_{+-}}{N_+} \tag{59}$$

$$\frac{1}{N_-} \sum_{i=1}^{N_-} \mu_-^\top \tilde{x}_i^{(-)} = \frac{1}{N_-} \sum_{i=1}^{N_{--}} \mu_-^\top \tilde{x}_i^{(-)} - \frac{N_{-+}}{N_-} \tag{60}$$

We will start by considering equation 59. Conditioned on the event that $\left| \frac{2}{N} \sum_{i=1}^{N/2} -\epsilon \right| \leq \frac{2\sqrt{\log(N/2)}}{N}$, which occurs with probability at least $1 - \frac{1}{N^2}$, we have that the right hand side is lower bounded by

$$\left( 1 - \epsilon - \frac{2\sqrt{\log(N/2)}}{N} \right) \left( \frac{1}{N_{++}} \sum_{i=1}^{N_{++}} \mu_+^\top \tilde{x}_i^{(+)} \right) - \epsilon - \frac{2\sqrt{\log(N/2)}}{N} \tag{61}$$

This is further lower bounded with probability at least $1 - \frac{1}{\alpha^2}$ by

$$1 - 2\epsilon - \frac{4\sqrt{\log N}}{N} - \frac{1}{\gamma} - \alpha \sqrt{\frac{8}{(N - \epsilon N - \sqrt{\log N})\gamma}} \tag{62}$$

as $(1-a)(1-b) \geq 1 - a - b$ for $0 \leq a, b$. By the same argument for equation 60, we can complete the proof. $\qquad\square$

**Theorem B.1** (Generalization Error with Noisy Samples). *Suppose we have a noisy dataset such that each sample has its labels flipped with probability $\epsilon$, with $0 \leq \epsilon \leq \frac{1}{2}$. Let $\tilde{x}_1^{(+)}, \tilde{x}_2^{(+)}, \ldots, \tilde{x}_{N_+}^{(+)}$ be the resulting set of samples that have labels corresponding to positive examples, and let $\tilde{x}_1^{(-)}, \tilde{x}_2^{(-)}, \ldots, \tilde{x}_{N_-}^{(-)}$ be the set of negative examples. Then, with probability at least $1 - \frac{2\mathcal{R}_0}{N - \epsilon N - \sqrt{\log N}} - \frac{2}{N^2}$, for $0 \leq \epsilon \leq \frac{1}{2} \left( 1 - \frac{1}{\gamma} - \cos \frac{\theta}{3} - \frac{4\sqrt{\log N}}{N} \right)$, for $0 < t \leq \frac{\sin(\theta/3)\tau}{4\beta^2 D}$, the population risk of the model is bounded as*

$$\mathcal{R}(\mathcal{P}) \leq \frac{\mathcal{R}_0}{\left( 1 - \sqrt{\delta\gamma} \left( \epsilon + \frac{2\sqrt{\log N}}{N} \right) \right)^2} \tag{63}$$

*where*

$$\mathcal{R}_0 = \frac{8}{\gamma \left( 1 - \frac{1}{\gamma} - \cos \frac{\theta}{3} \right)^2} \tag{64}$$

**Proof.** By Lemma B.5 and B.6, we have that with probability at least $1 - \frac{2\mathcal{R}_0}{N - \epsilon N - \sqrt{\log N}} - \frac{2}{N^2}$,

$$\frac{1}{N_+} \sum_{i=1}^{N_+} \mu_+^\top \tilde{x}_i^{(+)} \geq 1 - 2\epsilon - \frac{4\sqrt{\log N}}{N} - \frac{1}{\gamma} - \sqrt{\frac{8}{\delta\gamma}} \tag{65}$$

$$\frac{1}{N_-} \sum_{i=1}^{N_-} \mu_-^\top \tilde{x}_i^{(-)} \geq 1 - 2\epsilon - \frac{4\sqrt{\log N}}{N} - \frac{1}{\gamma} - \sqrt{\frac{8}{\delta\gamma}} \tag{66}$$

and as $\mathcal{R}_0 = \frac{8}{\gamma\left(1 - \frac{1}{\gamma} - \cos\frac{\theta}{3}\right)^2}$, we have that

$$\frac{1}{N_+} \sum_{i=1}^{N_+} \mu_+^\top \tilde{x}_i^{(+)} \geq 1 - 2\epsilon - \frac{4\sqrt{\log N}}{N} - \frac{1}{\gamma} - \left(1 - \frac{1}{\gamma} - \cos\frac{\theta}{3}\right) \tag{67}$$

$$\frac{1}{N_-} \sum_{i=1}^{N_-} \mu_-^\top \tilde{x}_i^{(-)} \geq 1 - 2\epsilon - \frac{4\sqrt{\log N}}{N} - \frac{1}{\gamma} - \left(1 - \frac{1}{\gamma} - \cos\frac{\theta}{3}\right) \tag{68}$$

and therefore

$$\frac{1}{N_+} \sum_{i=1}^{N_+} \mu_+^\top \tilde{x}_i^{(+)} \geq \cos\frac{\theta}{3} - 2\epsilon - \frac{4\sqrt{\log N}}{N} \tag{69}$$

$$\frac{1}{N_-} \sum_{i=1}^{N_-} \mu_-^\top \tilde{x}_i^{(-)} \geq \cos\frac{\theta}{3} - 2\epsilon - \frac{4\sqrt{\log N}}{N} \tag{70}$$

Let $\phi = \arccos\left(\cos\frac{\theta}{3} - 2\epsilon - \frac{4\sqrt{\log N}}{N}\right) - \frac{\theta}{3}$. By Lemma B.3, we know that for $0 < t \leq \frac{\sin(\theta/3)\tau}{4\beta^2 D}$, the boundary at time $t$ is within an angle of $\theta/3$ from the initial boundary. Then, as $\mu_+ - \mu_-$ is $\theta$ away from each of $\mu_+, \mu_-$, we know that any sample within an angle of $\theta/3 - \phi$ from the corresponding mean will be classified correctly. Since cosine is concave for angles between 0 and $\pi/2$, we have that $\cos(\theta/3 - \phi) \leq \cos\frac{\theta}{3} + 2\epsilon + \frac{4\sqrt{\log N}}{N}$. Then, we can guarantee that a new sample is classified correctly if its dot product with its corresponding mean is at least $\cos\frac{\theta}{3} + 2\epsilon + \frac{4\sqrt{\log N}}{N}$. For a new sample, by Lemma B.1, this occurs with probability at least

$$1 - \frac{8}{\gamma\left(1 - \frac{1}{\gamma} - \cos\frac{\theta}{3} - 2\epsilon - \frac{4\sqrt{\log N}}{N}\right)^2} \tag{71}$$

or

$$1 - \frac{\mathcal{R}_0}{\gamma\left(1 - \sqrt{\mathcal{R}_0\gamma}\left(2\epsilon - \frac{4\sqrt{\log N}}{N}\right)\right)^2} \tag{72}$$

and therefore the risk is upper bounded as

$$\mathcal{R}(\mathcal{P}) \leq \frac{\mathcal{R}_0}{\left(1 - \sqrt{\mathcal{R}_0\gamma}\left(\epsilon + \frac{2\sqrt{\log N}}{N}\right)\right)^2} \tag{73}$$

$\square$

**Theorem B.2 (Behavior of expected risk).** *Suppose we have a noisy dataset such that each sample has its label flipped with probability $\epsilon$. Then, for $0 \leq \epsilon \leq 1 - \frac{1}{\gamma} - \cos\frac{\theta}{3} - \frac{\sqrt{\log N}}{N}$ and $0 < t \leq \frac{\sin(\theta/3)\tau}{4\beta^2 D}$, the expected population risk of the model $\mathbb{E}_{\tilde{\mathcal{D}}_\epsilon}[\mathcal{R}(\mathcal{P})]$, averaged over the sampled noisy datasets $\tilde{\mathcal{D}}_\epsilon$, is bounded by*

$$\mathbb{E}_{\tilde{\mathcal{D}}_\epsilon}[\mathcal{R}(\mathcal{P})] \leq \frac{\mathcal{R}_0}{\left(1 - \sqrt{\mathcal{R}_0\gamma}\left(\epsilon + \frac{2\sqrt{\log N}}{N}\right)\right)^2} + \frac{2\mathcal{R}_0}{N - \epsilon N - \sqrt{\log N}} + \frac{2}{N^2}. \tag{74}$$

*Additionally, we have that for any $t$ and for any $\theta, \gamma$,*

$$\frac{d^2}{d\epsilon^2}\mathbb{E}_{\tilde{\mathcal{D}}_\epsilon}[\mathcal{R}(\mathcal{P})]\Big|_{\epsilon=1/2} = 0 \tag{75}$$

**Proof.** By Theorem B.1, we have that with probability at least $1 - \frac{2\mathcal{R}_0}{N - \epsilon N - \sqrt{\log N}} - \frac{2}{N^2}$,

$$\mathcal{R}(\mathcal{P}) \leq \frac{\mathcal{R}_0}{\left(1 - \sqrt{\mathcal{R}_0 \gamma}\left(\epsilon + \frac{2\sqrt{\log N}}{N}\right)\right)^2} \tag{76}$$

and that $\mathcal{R}(\mathcal{P})$ is always less than or equal to 1, so

$$\mathbb{E}_{\tilde{\mathcal{D}}_\epsilon}[\mathcal{R}(\mathcal{P})] \leq \frac{\mathcal{R}_0}{\left(1 - \sqrt{\mathcal{R}_0 \gamma}\left(\epsilon + \frac{2\sqrt{\log N}}{N}\right)\right)^2} + \frac{2\mathcal{R}_0}{N - \epsilon N - \sqrt{\log N}} + \frac{2}{N^2} \tag{77}$$

Now, we consider $\mathbb{E}_{\tilde{\mathcal{D}}_\epsilon}[\mathcal{R}(\mathcal{P})]$. Let $x_1, \ldots, x_N$ represent the sample embeddings and let $z_1, \ldots, z_N$ be $\mathrm{Ber}(\epsilon)$ variables that determine the label flipping. Then,

$$\mathbb{E}_{\tilde{\mathcal{D}}_\epsilon}[\mathcal{R}(\mathcal{P})] = \int_{\mathbb{R}^d} \cdots \int_{\mathbb{R}^d} \mathbb{E}_{z_1, \ldots, z_N}[\mathcal{R}(\mathcal{P})|x_1, \ldots, x_N]\nu(x_1, \ldots, x_N)dx_1 \ldots dx_N \tag{78}$$

where $\nu(x_1, \ldots, x_N)$ is the joint density of the sample embeddings. We can additionally expand $\mathbb{E}_{z_1, \ldots, z_N}[\mathcal{R}(\mathcal{P})|x_1, \ldots, x_N]$ as a sum over the $2^N$ possible $z_1, \ldots, z_N$ configurations. Since $\epsilon$ appears only within the sum and the sum is polynomial in $\epsilon$, we know that $\mathbb{E}_{\tilde{\mathcal{D}}_\epsilon}[\mathcal{R}(\mathcal{P})]$ is twice differentiable in $\epsilon$ as we can move $\frac{d^2}{d\epsilon^2}$ inside the integral and inside the sum. Now, we will show that

$$\mathbb{E}_{\tilde{\mathcal{D}}_\epsilon}[\mathcal{R}(\mathcal{P})]\big|_\epsilon = 1 - \mathbb{E}_{\tilde{\mathcal{D}}_\epsilon}[\mathcal{R}(\mathcal{P})]\big|_{1-\epsilon} \tag{79}$$

Since $\nu(x_1, \ldots, x_N)$ is independent of $\epsilon$, the above is true if for a given $x_1, \ldots, x_N$,

$$\mathbb{E}_{z_1, \ldots, z_N}[\mathcal{R}(\mathcal{P})|x_1, \ldots, x_N] = 1 - \mathbb{E}_{z_1', \ldots, z_N'}[\mathcal{R}(\mathcal{P})|x_1, \ldots, x_N] \tag{80}$$

where $z_1, \ldots, z_N \sim \mathrm{Ber}(\epsilon)$ and $z_1', \ldots, z_N' \sim \mathrm{Ber}(1 - \epsilon)$. The probability of sampling a given $z_1, \ldots, z_N$ is the same as sampling $z_1', \ldots, z_N'$ with the exact opposite set of labels being flipped. We know that the reward dynamics, for any sample $(x^*, y_w^*, y_l^*)$ and letting $r^*$ be its reward margin, follow

$$\tau \dot{r^*} = \frac{1}{N}\sum_{i=1}^N \beta^2 f'(r_i)(\mathbf{y}_w^* - \mathbf{y}_l^*)^\top(\tilde{\mathbf{y}}_{w,i} - \tilde{\mathbf{y}}_{l,i})g(x^*)^\top g(x_i) \tag{81}$$

Additionally, the reward dynamics for the training samples are the same for $z_1, \ldots, z_N$ and $z_1', \ldots, z_N'$, so the reward dynamics for any new sample are the exact opposite for $z_1, \ldots, z_N$ and $z_1', \ldots, z_N'$. This means that the resulting models have exact opposite predictions and therefore

$$\mathbb{E}_{z_1, \ldots, z_N}[\mathcal{R}(\mathcal{P})|x_1, \ldots, x_N] = 1 - \mathbb{E}_{z_1', \ldots, z_N'}[\mathcal{R}(\mathcal{P})|x_1, \ldots, x_N] \tag{82}$$

Now, since we know that

$$\mathbb{E}_{\tilde{\mathcal{D}}_\epsilon}[\mathcal{R}(\mathcal{P})]\big|_\epsilon = 1 - \mathbb{E}_{\tilde{\mathcal{D}}_\epsilon}[\mathcal{R}(\mathcal{P})]\big|_{1-\epsilon} \tag{83}$$

We can apply $\frac{d^2}{d\epsilon^2}$ to both sides and we have that

$$\frac{d^2}{d\epsilon^2}\mathbb{E}_{\tilde{\mathcal{D}}_\epsilon}[\mathcal{R}(\mathcal{P})]\bigg|_\epsilon = -\frac{d^2}{d\epsilon^2}\mathbb{E}_{\tilde{\mathcal{D}}_\epsilon}[\mathcal{R}(\mathcal{P})]\bigg|_{1-\epsilon} \tag{84}$$

and at $\epsilon = 1/2$, this is only possible if

$$\frac{d^2}{d\epsilon^2}\mathbb{E}_{\tilde{\mathcal{D}}_\epsilon}[\mathcal{R}(\mathcal{P})]\bigg|_{\epsilon=1/2} = 0 \tag{85}$$

$\square$

# C    VMF DISTRIBUTION VERIFICATION

We verify that the embeddings from real-world models and datasets exhibit key characteristics of the vMF distribution. We use the Anthropic Persona dataset (Perez et al., 2022) which consists of a diverse set of personas. For each persona, there is a collection of 500 statements that align with the persona, and 500 statements that misalign with the persona. These samples can be viewed as positive and negative samples, respectively. All embeddings are collected after RMSNorm has been applied. We collect the norm of the final layer embedding at the end of each statement and calculate both the average norm and the variance across all samples. As depicted in the first two rows of Table 2, the embeddings consistently show a similar norm with small variance, approximately conforming to the vMF distribution. Additionally, for every persona, we compute the mean embedding of the positive and negative samples, and calculate the cosine similarity between each sample and its corresponding mean. We then average the cosine similarity for the positive samples and the negative samples, and compile these averages across all personas. The results, shown in the last two rows of Table 2, demonstrate high average cosine similarities with minimal variance. This suggests that the embeddings are concentrated around their respective means across personas, supporting the presence of the vMF distribution in a real-world dataset, aligning with our theoretical setup.

Table 2: Verification of vMF distribution.

| | |
|---|---|
| Average norm | 140.3 |
| Norm Variance | 1.618 |
| Average cosine | 0.9876 |
| Cosine Variance | 1.106e-5 |

