# OpenReview forum: "Understanding Generalization of Preference Optimization Under Noisy Feedback"
_ICLR.cc/2025/Conference — Submitted to ICLR 2025_

### Official Review · Reviewer_9Dyv · 2024-11-01

**Soundness:** 3
**Presentation:** 1
**Contribution:** 2
**Rating:** 3
**Confidence:** 3

**Summary:**

This paper studies the problem of preference optimization about the impact of noisy feedback. The key theoretical observation is that a formula of the risk upper bound that increases with extent of noise. When the noise rate approaches 1/2, the expected risk transitions grows at a linear rate. The theory also reveals that stronger concentration, more samples, and contrasting
directions for positive and negative samples yields tighter bounds and slower degradation in
accuracy as the noise rate increases. Empirical evaluations are  done to support the theoretical findings.

**Strengths:**

1.The studied policy optimization class is general, including many commonly used algorithms.

2.The proposed generalization guarantees can reflect the empirical observation, as verified in the experiment part.

3.The analysis of this problem is very novel.

**Weaknesses:**

1.It is unclear why the result of Lemma 3.1 is important. In my understanding, it is very technical and serves for the proof of the main results. The writing lacks intuition and presenting the proof is not helpful for understanding.

2.Theorems 3.1 and 3.2 hold only when $t$ is not very large. What will happen after that? When $t$ is close to 0, it means that even without any training, but the risk has almost the same guarantee. Does that make sense?

3.The presentation of formulas should be improved. The $f_\theta$ in Line 207 is different from the $f$ defined in equation (9)? What role does it play in this setting? What are the "one hot vectors of the token" defined in Line 230? What’s the dimension of this vector? And how do you get equation (11)? In Theorem 3.1, what are $\tau$ and $D$?

4.I don’t quite get the message in the experiment part. In Section 4.1, the theoretical fits seem like linear? But the theoretical result is not? How do you choose $c$ in the graph? It is very strange to claim the approximation is good without specifying the parameters. Moreover, It says the approximation is good for $\epsilon < 0.5$, but the experiment is only done for $\epsilon < 0.4$? In Section 4.2, still I don’t get how the empirical line is drawn, and why the approximation can reflect anything. Why can’t we just treat the trend as linear and use a linear line to approximate it instead of the proved $1/(1-c\epsilon)^2$?

In general, although there are many parts to be clarified, I can almost understand the theoretical results. However, the experiment part should be further explained and the paper should be further polished to convey the message clearly.

**Questions:**

See Weaknesses

---

### Official Review · Reviewer_4BQX · 2024-11-01

**Soundness:** 3
**Presentation:** 2
**Contribution:** 2
**Rating:** 5
**Confidence:** 4

**Summary:**

The authors studied the impact of noise in feedback on the alignment of large language models (LLMs) through preference optimization. In particular, they considered a fairly general alignment loss which includes DPO, IPO, and SLiC. Moreover, they assumed that the "feature backbone" $g(x)$ of the model (the term is not clearly defined in the paper but it seems that it refers to the last layer of the model) is fixed (which I think it means that weights of the model are frozen) and only the "unembbeding matrix" $W$ is learnable. They modeled the output of the model by softmax$(Wg(x))$ and analyzed the dynamic of gradient flow over $W$ under some assumptions on the distribution of the input (the von Mises-Fisher (vMF) distribution). The authors defined a notion of risk for preference learning (Definition 3.1) and they provided upper bounds on the expected population risk (Theorem 3.2) through the gradient flow analysis (under some assumption on the significance of noise (which is modeled by a Bernoulli RV with parameter $\epsilon$) in the feedback and the run time $t$). Based on their analysis, they observed that the expected risk is bounded by $1/(1-c\epsilon)^2$ when $\epsilon$ is sufficiently below $1/2$ and then decreases linearly with $\epsilon$ when $\epsilon$ gets close to $1/2$. They verified their analysis on some synthetic data and also a real setting.

**Strengths:**

Originality: The authors studied the impact of noise in the feedback on the performance of preference optimization in LLMs. To me, the dependency of performance on the level of noise (which is characterized in this paper) is new in the field.

Quality/Signgiance: The result is derived based on heavy assumptions on the training of LLM and the distribution of input. Therefore, it may not capture the exact dependency of the performance on $\epsilon$.

Clarity: Some of the explanations are missing in the paper and assumptions are not mentioned in a separate item that we can refer to. Therefore, in this respect, I do not see any remarkable strength in the paper.

**Weaknesses:**

Presentation: The paper is somehow readable until the end of Section 2 but then it is hard to follow (sometimes the notations/terms are not clearly identified). Moreover, the main assumptions in the paper should be stated in an assumption item and then be referred to in the statements of theorems. Moreover, the abstract is somehow misleading as these assumptions are not mentioned there.

Justifications for the assumptions: For some of the assumptions in the theorems, it is good to add some justifications on why these are reasonable assumptions.

Implications of the analysis: It is good to mention the implication of this analysis in preference optimization (For instance, how this analysis can be helpful in designing a better alignment method). Moreover, it is hard to characterize the boundary between two regimes of $1/(1-c\epsilon)^2$ and the linear one.

**Questions:**

1. Why is a binary loss suitable for defining risk in Definition 3.1? For instance, why not considering the reward margin itself as the risk?

2. In line 208, the authors used the term "feature backbone" without explicitly defining it. As one of the key assumptions in the analysis is given here, I suggest clearly mentioning the assumption as a separate item and refer it in the statement of theorems. Otherwise, it is hard to know which assumptions are made in each theorem.

3. I think it is good to talk a little bit about gradient flow why it is considered and the notations used there (such as the learning rate $\tau$). The analysis is also limited to gradient-descent-like updates and may not capture the dynamics of other optimization algorithms (such as AdamW) which are often used in the alignment of LLMs.

4. Please use an assumption item for the assumption on the input distribution. Moreover, I am not completely sure that the statement in line 258 has been observed in the literature. So, please give a citation for it. The experiments in Appendix C are limited to one dataset and it is not clear on what model has been tested.

5. In the statement of Theorems 3.1 and 3.2, the run time $t$ should be bounded by a factor of learning $\tau$. So, if I am not mistaken, it is limited to one step in the discretized version of the gradient descent algorithm.

6. It is good to characterize the boundary between two regimes of $1/(1-c\epsilon)^2$ and the linear one.

7. In the experiments on real data, it is asserted that the linear dependency observed as the original data is very noisy in itself. I suggest annotating the dataset with a third-party model (such as chatGPT4) or using available datasets (such as Alpacafarm) and rerun the experiments with a data that is less noisy.

---

### Official Review · Reviewer_svts · 2024-11-01

**Soundness:** 2
**Presentation:** 2
**Contribution:** 2
**Rating:** 5
**Confidence:** 4

**Summary:**

The paper studies preference optimization with noisy preference
feedback. Theoretically, the paper analyzes a fairly broad family of
algorithms under softmax-linear parametrization of the language model
policy. Experimentally, the paper mostly focuses on DPO loss both in
controlled settings (but also considers some other losses) and on
standard RLHF benchmarks. The main technical tool is a notion of
reward margin, essentially the implicit reward difference between
prefered and dis-preferred responses and the main concept insight is
that stronger "concentration" (less variance of the data), more
samples, and increasing the "angle" between positive and negative
directions can improve performance.

Comments:

1. I have a basic issue with the premise of this paper. The original
derivation of DPO (and these related losses) _already_ incorporates
noise in the preference data. The data is assumed to be generated by
the Bradley-Terry model, in which preference pairs appear via
P(y_1>y_2) \propto \sigma(r(y_1) - r(y_2)). This is already noisy! And
the BT model can nearly model the "massart" noise model considered
here, all "true positives" have reward r_1 and all true negatives have
reward r_2 such that \sigma(r_1 - r_2) = 1-eps. So why not just
reframe the paper this way?

2. The setup for the analysis is highly specialized and I feel this is
rather limiting. Why is the vMF distribution natural or reasonable?
Why not use a more general noise model, like the original BT? Why this
specific policy parametrization? (I am ok with using softmax linear,
but I think this should be at the token level over the responses,
which I don't think is what is happening in the paper)

3. I am not particularly convinced by the controlled experiments. The
settings nearly perfectly matches the one in the theoretical
analysis. Thus the only conclusion one can make is, roughly, the
theorem is correct. But I do not know if the controlled experiments
have much bearing on practice.

4. The real-world experiment is more interesting, but I am not sure if
we should view this as validation of the theory, because we do not see
the 1/(1-x)^2 type behavior very clearly. Instead we see linear
behavior, which could be presumably explained by many other models. As
just a concrete example, in noisy classification (PAC learning with
classification noise) we expect that the error rate of ERM scales
linearly with the noise level, i.e., we expect err(ERM) \leq OPT +
poly(1/n) and OPT will incur error rate equal to the noise level. So
this is an equally valid, and much more general/simple/convincing,
explanation for the experimental results?  I get the stated reason for
why we do not see the 1/(1-x)^2 behavior, but the point remains that
no other competing hypothesese have not be convincingly ruled out.

4. I looked at the prior work Im-Li-2024a and I noticed that
nontrivial portions of the text seem to be copied verbatim. I am not
sure if this is problematic or not...

Overall: It's quite possible I am missing something and I'm open to
having my mind changed about this. But, I feel the paper should just
work in the standard BT model, should consider a more general setup
for the analysis, and needs to have a more convincing experimental
section. As it stands, I feel the paper is below the bar for ICLR.

**Strengths:**

see above

**Weaknesses:**

see above

**Questions:**

see above

---

### Official Review · Reviewer_RBPH · 2024-11-02

**Soundness:** 2
**Presentation:** 1
**Contribution:** 2
**Rating:** 3
**Confidence:** 4

**Summary:**

This paper explores the generalization of preference optimization in large language models under noisy human feedback, providing theoretical guarantees and empirical validation in both synthetic and real-world settings.

**Strengths:**

The paper offers theoretical insights into noisy preference optimization, extends to a broad class of methods, and validates its findings through empirical analysis on controlled and real-world datasets.

**Weaknesses:**

1. Identical text to the previous work [1]： The text has significant overlap with prior work by Im and Li [1]. The authors should clarify how their contributions differ, especially when the preliminary section is almost identical to those in [1], and the equations are used verbatim.

    [1] Shawn Im and Yixuan Li. On the generalization of preference learning with dpo

2. The problem setting is artificial. The prompt embedding is assumed from a vMF distribution. In the extreme case ($\gamma \rightarrow +\infty$), the problem becomes that there are only two possible prompts $\mu^+$ and $\mu^-$. And the preference learning is conducted only for the two prompts.

3. The theoretical framework applies solely to single-token preference learning, which might limit its applicability, as LLM preference learning typically involves response sequences rather than isolated tokens.

4. The setting is ambiguous and the mathematical treatment is poorly presented:
   - There is no statement on the assumption of the ground-truth preference model. Is it assumed that noiseless human preference strictly obey an underlying reward model? If so it should be clearly stated. Otherwise, a circular preference relation $A \succ B \succ C \succ A$ will guarantee the population risk can not reach $0$, while Theorem 3.1 here implies it can be achieved under any circumstances.
   - The data generation procedure is poorly stated and artificial. Why there are two and only two clusters of prompt? And why must they be equally distributed. Additionally, given the sample from $\mathcal{D}_+$ or $\mathcal{D}_-$, how is the chosen and rejected token generated? What is their distribution and what preference model do they follow?

**Questions:**

See Weaknesses.

1. Can you clarify your choice of vMF distribution for prompt embedding and whether alternative distributions were considered for modeling prompt variance?
2. Are there specific empirical results to further substantiate the claim that modern LLM embeddings indeed follow a vMF distribution?

---

### Meta-Review · Area_Chair_RtKV · 2024-12-21

**Metareview:**

This paper addresses the impact of noisy feedback on preference optimization for aligning large language models (LLMs) with human preferences. While most existing work assumes noise-free feedback, this paper provides generalization guarantees under noisy conditions. The authors establish guarantees for noisy preference learning across various optimization losses (e.g., DPO, IPO, SLiC) and present a general model describing how generalization decays with the noise rate. Empirical results on contemporary LLMs validate the practical relevance of these findings. The reviewers raised many critical questions, but the authors did not provide any responses. As a result, I recommend rejection.

**Additional Comments On Reviewer Discussion:**

The authors did not provide rebuttal.

---

### Decision · Program_Chairs · 2025-01-22

Reject